# Synthesis of Novel Arylhydrazones Bearing 8-Trifluoromethyl Quinoline: Crystal Insights, Larvicidal Activity, ADMET Predictions, and Molecular Docking Studies

**DOI:** 10.3390/ph18121804

**Published:** 2025-11-26

**Authors:** Sukumar Kotyan, Shankaranahalli N. Chandana, Doddabasavanahalli P. Ganesha, Banavase N. Lakshminarayana, Nefisath Pandikatte, Pran Kishore Deb, Manik Ghosh, Raquel M. Gleiser, Mohamad Fawzi Mahomoodally, Sukainh Aiaysh Alherz, Mohamed A. Morsy, Hany Ezzat Khalil, Mahesh Attimarad, Sreeharsha Nagaraja, Rashed M. Almuqbil, Abdulmalek Ahmed Balgoname, Bandar E. Al-Dhubiab, Afzal Haq Asif, Katharigatta N. Venugopala, Jagadeesh Prasad Dasappa

**Affiliations:** 1Department of Chemistry, Mangalore University, Mangalagangotri 574199, Karnataka, India; sukumarkotyan8793@gmail.com; 2Department of Physics, Rajeev Institute of Technology, Hassan 573201, Karnataka, India; snc.rit@gmail.com; 3Department of Physics, Malnad College of Engineering, Affiliated to Visvesvaraya Technological University, Hassan 573201, Karnataka, India; ganeshaphysics@gmail.com; 4Research Center, Department of Physics, Adichunchanagiri Institute of Technology, Affiliated to Visvesvaraya Technological University, Jyothinagara, Chikkamagaluru 577102, Karnataka, India; 5Department of PG Studies and Research in Chemistry, Sri Dharmasthala Manjunatheshwara College (Autonomous), Ujire 574240, Karnataka, India; nafeesath@sdmcujire.in; 6Department of Pharmaceutical Sciences and Technology, Birla Institute of Technology (BIT), Mesra 835215, Jharkhand, India; prankishoredeb@bitmesra.ac.in (P.K.D.); manik@bitmesra.ac.in (M.G.); 7CREAN-IMBIV (UNC-CONICET), Av. Valparaiso s.n., and FCEFyN, Universidad Nacional de Cordoba, Av. V. Sarsfield 299, Cordoba 5000, Argentina; raquel.gleiser@unc.edu.ar; 8Institute of Research and Development, Duy Tan University, Da Nang 550000, Vietnam; mohamadfawzimahomoodally@duytan.edu.vn; 9School of Engineering & Technology, Duy Tan University, Da Nang 550000, Vietnam; 10Laboratory of Natural Products and Medicinal Chemistry (LNPMC), Center for Global Health Research, Saveetha Medical College and Hospital, Saveetha Institute of Medical and Technical Sciences (SIMATS), Thandalam, Chennai 602105, Tamil Nadu, India; 11Centre of Excellence for Pharmaceutical Sciences (Pharmacen), North West University, Potchefstroom 2520, South Africa; 12Department of Health Sciences, Faculty of Medicine and Health Sciences, University of Mauritius, Reduit 80837, Mauritius; 13Department of Pharmaceutical Sciences, College of Clinical Pharmacy, King Faisal University, Al-Ahsa 31982, Saudi Arabia; salharz@kfu.edu.sa (S.A.A.); momorsy@kfu.edu.sa (M.A.M.); heahmed@kfu.edu.sa (H.E.K.); mattimarad@kfu.edu.sa (M.A.); sharsha@kfu.edu.sa (S.N.); ralmuqbil@kfu.edu.sa (R.M.A.); abalgoname@kfu.edu.sa (A.A.B.); baldhubiab@kfu.edu.sa (B.E.A.-D.); 14Department of Pharmacy Practice, College of Clinical Pharmacy, King Faisal University, Al-Ahsa 31982, Saudi Arabia; ahasif@kfu.edu.sa; 15Department of Biotechnology and Food Science, Faculty of Applied Sciences, Durban University of Technology, Durban 4001, South Africa

**Keywords:** 8-trifluoromethyl quinolone, hydrazine, larvicidal activity, XRD, DFT, *Anopheles arabiensis*

## Abstract

**Background/Objectives**: Vector-borne diseases like malaria remain a major global health concern, worsened by insecticide resistance in mosquito populations. Quinoline-based compounds have been extensively studied for their pharmacological effects, including antimalarial and larvicidal properties. Modifying quinoline structures with hydrazone groups may enhance their biological activity and physicochemical properties. This study reports the synthesis, structural characterization, and larvicidal testing of a new series of aryl hydrazones (**6a**–**i**) derived from 8-trifluoromethyl quinoline. **Methods**: Compounds **6a**–**i** were prepared via condensation reactions and characterized using ^1^H NMR, ^19^F-NMR, ^13^C NMR, and HRMS techniques. Their larvicidal activity was tested against *Anopheles arabiensis*. Single-crystal X-ray diffraction (XRD) was performed on compound **6d** to determine its three-dimensional structure. Hirshfeld surface analysis, fingerprint plots, and interaction energy calculations (HF/3-21G) were used to examine intermolecular interactions. Quantum chemical parameters were computed using density functional theory (DFT). Molecular docking studies were performed for the synthesized compounds **6a**–**i** against the target acetylcholinesterase from the malaria vector (6ARY). In silico ADMET properties were also calculated to evaluate the drug-likeness of all the tested compounds. **Results**: Compound **6a** showed the highest larvicidal activity, causing significant mortality in *Anopheles arabiensis* larvae. Single-crystal XRD analysis of **6d** revealed a monoclinic crystal system with space group P2_1_/c, stabilized by N–H···N intermolecular hydrogen bonds. Hirshfeld analysis identified H···H (22.0%) and C···H (12.1%) interactions as key contributors to molecular packing. Density functional theory results indicated a favorable HOMO–LUMO energy gap, supporting molecular stability and good electronic distribution. The most active compounds, **6a** and **6d**, also showed strong binding interactions with the target protein 6ARY and satisfactory ADMET properties. The BOILED-Egg model is a powerful tool for predicting both blood–brain barrier (BBB) and gastrointestinal permeation by calculating the lipophilicity and polarity of the reported compounds **6a**–**i**. **Conclusions**: The synthesized arylhydrazone derivatives demonstrated promising larvicidal activity. Combined crystallographic and computational studies support their structural stability and suitability for further development as eco-friendly bioactive agents in malaria vector control.

## 1. Introduction

Heterocyclic compounds are widespread in nature, as they exhibit various biological properties and are used in clinical applications [1]. In recent years, it has been found that the stability of the quinoline derivatives has shown particular importance in the field of the medicinal industry in a variety of bioactive skeletons, creating novel potential therapeutic medicines (Figure 1). Quinoline is found to act as a potential therapeutic agent for malaria [2], cancer [3], and bacterial diseases [4]. Further, the quinoline compounds are considered to exhibit strong potential for diverse biological activities, such as anti-tubercular [5], antifungal [6], antibacterial [7], anticonvulsant [8], analgesic and anti-inflammatory [9,10], antiallergic [11], anti-amoebic [12], antianxiety [13], larvicidal [14], anticancer [15], antiviral [16] and antimalarial [17] properties. Undoubtedly, quinoline has become the most essential drug in the anti-infective chemotherapy field [18]. Recent studies have increasingly focused on fluorinated compounds, as they exhibit significant pharmaceutical activities. Replacing hydrogen atom(s) with fluorine has been found to enhance the biological activities in certain compounds [19,20]. Trifluoromethyl-substituted compounds have been documented to display diverse biological activities, including herbicides [21,22], antitumor [23], fungicides [24], and platelet aggregation inhibitors [25]. Similarly, the structural versatility of hydrazones, which contain the –CH=N–NH– group responsible for their reactivity, forms a key class of organic compounds. Hydrazone derivatives exhibit various biological activities, such as larvicidal [26], antimicrobial [27], analgesic, antioxidant [28], and anti-inflammatory [29] activities.

A key factor in the prevention and eradication of multiple vector-borne diseases, such as malaria, has been the implementation of vector control measures through the application of insecticides to eliminate the vector as the primary method [30]. However, the rise in insecticide resistance (besides environmental and health concerns from some compounds) poses a significant threat to the effectiveness of these interventions [31]. Therefore, there exists an immediate requirement to explore alternative prevention strategies, including the development of new insecticidal compounds [32]. Quinoline-scaffolded compounds are extensively investigated in experimental and theoretical studies in nonlinear optics (NLO), a crucial field of study, due to their low dielectric constants and ultrafast, broadband electronic responses, making them promising for optoelectronic technologies [33,34]. The synthesis of quinoline derivatives has gained significant interest in organic chemistry because of their biological and pharmacological uses mentioned earlier. Based on these findings, this study reports the synthesis of benzylidenehydrazinyl derivatives of 8-trifluoromethyl quinolone, characterized by ^1^H NMR, ^19^F NMR, ^13^C NMR, high-resolution mass spectrometry (HRMS), X-ray crystallography, Hirshfeld surface analysis, and density functional theory (DFT) studies, aimed at developing new larvicidal agents.

## 2. Results and Discussion

### 2.1. Chemistry

The intermediate 4-hydrazinyl-8-(trifluoromethyl)quinoline (**5**) was synthesized by the reaction of 4-chloro-8-trifluoromethyl quinoline (**4**) with hydrazine hydrate in ethanol using the reflux method [35]. The compound (**5**) was treated with different substituted benzaldehydes with ethanol at reflux to obtain (*E*)-4-(2-benzylidenehydrazinyl)-8-(trifluoromethyl)quinolone derivatives (**6a**–**i**) as illustrated in Figure 1 [35]. The ^1^H-NMR spectrum of compound (**5**) showed a multiplet at δ 7.47 to 8.73 ppm due to quinoline and NH protons. A broad singlet of two protons at δ 4.50 ppm was due to NH_2_ protons. The molecular ion peak at *m*/*z* 227.18 was consistent with the molecular formula C_10_H_8_F_3_N_3,_ confirming the identity of compound **5**.

The ^1^H-NMR spectrum of compound **6d** displayed a singlet at δ 8.33 ppm, attributed to the -N=CH- proton. A singlet at δ 11.31 ppm was due to the NH proton. δ at 7.51–8.68 ppm was ascertained to be 8 protons of trifluoromethyl quinoline and benzylidene moiety, thus confirming the formation of the desired product. The ^19^F NMR spectrum of compound (*E*)-4-(2-(3-bromo-4-fluorobenzylidene)hydrazinyl)-8-(trifluoromethyl)quinoline (**6a**) displays two distinct singlets at δ –58.7 ppm and δ –107.01 ppm, representing the trifluoromethyl (CF_3_) group and the aromatic fluorine atom, respectively. The absence of additional splitting and the clear baseline confirm the presence of two chemically different fluorine environments, consistent with the proposed structure of (**6a**). The ^19^F NMR spectra of the target compounds **6b**–**g** and **6i** show a single sharp singlet at –58.7 ppm, indicating the trifluoromethyl (CF_3_) group. The presence of only one fluorine resonance and the lack of extra signals confirm the integrity of the CF_3_ moiety and align with the proposed structures. The molecular ion peaks at *m*/*z* 424.0266 [M+1] obtained for the targeted compound **6d** were in concurrence with the molecular formula C_18_H_13_BrF_3_N_3_O (Appendix A). Physicochemical characteristics of the series of compounds (**6a**–**i**) are given in Table 1.

### 2.2. Single-Crystal Structural Analysis

The single-crystal X-ray diffraction (XRD) studies show that the crystal of the title compound (**6d**) crystallizes in the monoclinic crystal system with the space group P2_1_/c having cell parameters *a =* 10.0716(7) Å, *b* = 14.5234(8) Å, *c* = 13.0461(7) Å, and β = 110.745°. Table 2 shows the crystal structure and refinement details. The methoxy group at C15 adopts an almost coplanar orientation with the benzene ring, as indicated by the torsion angle of 169.6(4)° for the atom C16-C15-O1-C18. Furthermore, it is observed that the structure displays intermolecular interactions of the type N2–H2…N1 and a weak C3-H3…F3 intramolecular interaction between molecules (Table 3). The structure exhibits weak π-π interactions between ring centroids of the benzene and quinoline rings (Table 4), linking the molecules into one-dimensional chains along the *a*-axis, contributing to overall crystal packing, as shown in Figure 2. The bond lengths, bond angles, and torsion angles of the compound **6d** are summarized in Table 5, Table 6 and Table 7, respectively. These structural parameters closely match those reported for related compounds in the literature [36,37]. The optimized geometrical parameters (Table 5 and Table 6, respectively), such as bond lengths and bond angles, show good correlation with experimental data (crystal structure) determined by XRD study. The molecular structure is composed of trifluoromethyl quinoline and bromo methoxybenzylidene rings bridged by a hydrazide chain with the chemical formula C_18_H_13_BrF_3_N_3_O. The thermal ellipsoid plot (ORTEP) of the molecule’s illustration drawn at a 50% probability is represented in Figure 3. The dihedral angle of 12.12(7)° is observed between the mean planes of the quinoline (N1/C1/C2/C3/C4/C5/C6/C7/C8/C9) and benzene (C12/C13/C14/C15/C16/C17) rings, which confirms that the moieties are in an *axial* orientation. The trifluoromethyl group is within the mean plane of the quinoline ring, which is conformed by torsion angle values of −2.5(6)° and 179.8(5)° defined by the atoms N1-C1-C2-C10 and C4-C3-C2-C10, respectively. The quinoline and phenyl rings lie in the same plane as the hydrazine moiety, as evidenced by the near-planar torsion angles of −174.4(4)° for the C7-N2-N3-C11 atoms and −179.7(4)° for the N2-N3-C11-C12 atoms [38].

The bond lengths and bond angles around the quinoline moiety and arylhydrazones are comparable to those of previously reported compounds [35]. It is also observed that the present compound **6d** exhibits both intermolecular interactions and pi-pi stacking, which are also seen in the earlier reported compound [35].

### 2.3. Computation Analyses

#### 2.3.1. Hirshfeld Surface Calculations

The d_norm_ plot was mapped with color scales ranging from −0.1290 a.u. (red) to 2.0311 a.u. (blue), respectively. The dark-red spot inside the contour on the d_norm_ surface arises as a result of the N2-H2…N1 hydrogen bond (Figure 4 and Table 5). As seen in Figure 5, the blue regions on the Hirshfeld surfaces mapped over electrostatic potential indicate an area of positive donor potential, while the bright-red areas at N1 (nitrogen) suggest a strongly electrostatic negative acceptor potential. Figure 6 shows the two-dimensional fingerprint plots, and Figure 7 displays the percentage of the contribution of overall interactions within a molecule. Among the intermolecular interactions, H…H contacts contributed the most with 22.0% of the total Hirshfeld surface, appearing as a pair of blue-colored blunt spikes directed towards the lower left region within the range 1.16 Å < (d_e_ + d_i_) < 1.19 Å. A pair of C…H/H…C interatomic contacts represented the next major contribution of 12.1%, which appears in the form of blue-colored wings around the region of 1.28 Å < (d_e_ + d_i_) < 1.83 Å. In the range of 1.3 Å (d_e_ + d_i_) < 1.92 Å, the Br…H/H…Br pair of interatomic interactions accounted for 11.4% of the contribution, which was represented by two identical fish shapes. With a contribution of 7.2%, the C…C intermolecular interactions are represented by an arrowhead of almost equal length that merges in the 1.75 Å < (d_e_ + d_i_) < 1.76 Å area. Spikes are displayed for the N…H/H…N (1.0 Å < (d_e_ + d_i_) < 1.4 Å) and O…H/H…O (1.12 Å < (d_e_ + d_i_) < 1.45 Å) pairs of interatomic interactions, which contributed 6.8% and 4.0%, respectively. A two-headed arrow representing the Br…C/C…Br contact with a contribution of 3.6% is displayed in the 1.72 Å (d_e_ + d_i_) < 1.96 Å region. C…F/F…C, N…C/C…N, O…C/C…O, and Br…F/F…Br all contribute the least to the surface’s overall area.

#### 2.3.2. Three-Dimensional Interaction Energy Frameworks

Figure 8 illustrates the interaction energies for molecule **6d** determined by building a cluster of molecules having a radius of 3.8 Å. Figure 9 shows the solid cylinders of size 150, with a threshold energy of 10 kJ/mol for the compound **6d**, displaying the Coulomb interaction energy (red), dispersion energy (green), and total interaction energy (blue) along the crystallographic *b*-axis. The breadth of the cylinders in energy frameworks provides information on the nature of various interactions and is proportional to the strength of the interactions. The framework’s cylinders can be made smaller or larger by adjusting the total scale factor Table 3.

The intermolecular N2-H2…N1 interactions are observed in a purple-colored molecule associated with a centroid distance of R = 7.54 Å, having a total interaction energy of E_tot_ = −65.1 kJ/mol. Table 8 provides information regarding the overall interaction energy linked to each molecule of a cluster. The dispersion energy outweighed the electrostatic energy, according to the final data.

#### 2.3.3. Molecular Electrostatic Potential (MEP)

The MEP is calculated using the optimized DFT/B3LYP method with the 6-311++G(d,p) basis set in the gas phase. MEP maps are extremely useful in predicting the behavior of organic molecules, which is used to visualize the electropositive and electronegative regions of the molecule. Figure 10 shows the MEP surface plot of the molecule (**6d**), ranging between −7.742 × 10^−2^ a.u. and +7.742 × 10^−2^ a.u. The red region represents the electrophilic, and the blue region indicates the nucleophilic attractions of the molecule. It can be observed that the N1 atom over the nitrogen is in the negative region, predicting that the N1 atom acts as an acceptor, which reveals the intermolecular interaction, confirming the N2-H2…N1 intermolecular interaction as shown in Table 3.

#### 2.3.4. Frontier Molecular Orbitals (HOMO–LUMO Energy)

Electronic properties such as highest occupied molecular orbital (HOMO)–lowest unoccupied molecular orbital (LUMO) energy (HOMO–LUMO energy), ionization energy (I), electron affinity (A), electronegativity (χ), chemical potential (μ), global hardness (η), global softness (s), and electrophilicity index (ω) are crucial to study the stability and reactivity of the compound. The DFT/B3LYP method is used to study the electronic properties with a basis set 6-311++G(d,p) in the gas phase. The HOMO–LUMO energy gap is illustrated in Figure 11, where the distribution of energy levels for both the HOMO and LUMO energies is primarily concentrated across the quinoline ring. Table 9 presents the global reactive parameters that have been calculated. It is found that a narrow energy gap is observed for molecule **6d**, confirming that it is a soft molecule with higher reactivity.

### 2.4. Computational Studies

The title compounds **6a**–**i** were further investigated to study their binding affinity toward the malaria vector using molecular docking simulations. The 6ARY-**6a** and 6ARY-**6i** complexes showed the highest binding affinity, with values of –10.4 kcal/mol for both complexes. All the complexes 6ARY-**6b**, 6ARY-**6c**, 6ARY-**6d**, 6ARY-**6e**, 6ARY-**6f**, 6ARY-**6g**, and 6ARY-**6h** demonstrated higher binding scores compared to those of the previously reported complexes [39]. The binding scores and docking interactions with amino acids are detailed in Table 10. The 2D interaction plot of the protein–ligand complex is illustrated in Figure 12, highlighting the interactions between the ligand and the residues.

### 2.5. ADMET Properties

Appendix A summarizes the ADME properties of all the designed compounds, offering insights into their potential behavior within the human body. Molar refractivity values ranged from 91.88 to 106.03, within the standard range of 30–140, indicating suitable molecular polarity. Drug-related ADMET properties, such as lipophilicity and solubility, were assessed using iLOGP and SILICOS-IT, and all the designed compounds show acceptable values. Water solubility, which is crucial for absorption and distribution, exhibits favorable log S values ranging from −5.12 to −6.79, indicating moderate to complete solubility for all compounds. High gastrointestinal (GI) absorption, ranging from 88.23% to 95.172%, suggests promising bioavailability. Human skin permeability is an essential parameter for understanding compound absorption, falling within the acceptable range of −4.92 to −5.70 cm/s. All tested compounds were found not to cross the BBB, and CNS permeability remained negative, with log PS values between −1.11 and −2.132. An evaluation of all compounds revealed acceptable cytochrome P450 (CYP) enzyme inhibition profiles, though minor deviations were observed for CYP1A2. Drug-likeness was assessed using multiple filters: while all compounds complied with Lipinski’s five rules, minor deviations were noted with the criteria of Ghose, Egan, and Muegge. Potential structural alerts for false-positive biological activity were analyzed using PAINS and Brenk filters, which revealed minor violations but no significant issues. The BOILED-Egg model (Figure 13) shows that the dot in the white region indicates that compounds 6a-6i are predicted to have potential absorption in the gastrointestinal (GI) tract. The red point (PGP-) signifies the ability to stay in the brain or the GI tract.

### 2.6. Toxicity Studies

The toxicity profiles of all the designed molecules (**6a**–**i**) were evaluated using the pkCSM web server [40] and SwissADME [41], as detailed in Table 11. Mutagenic potential was assessed via the AMES test, which yielded negative results for compounds **6b**, **6d**, and **6g**; the other compounds lack AMES-related toxicity. None of the compounds displayed hERG I inhibition, and all avoided hERG II inhibition, thereby reducing potential risks to the cardiovascular system. For rat toxicity, the Oral Acute Toxicity (LD50) values range from 2.657 to 3.347 mol/kg, and the Oral Chronic Toxicity (LOAEL) values range from 0.663 to 1.613 Log mg/kg·bw/day, indicating a generally safe profile. Although all compounds showed some level of hepatotoxicity, this risk can be mitigated by dose adjustments, structural modifications, or co-administration of protective agents. None of the compounds exhibits skin sensitization, and toxicity values for Tetrahymena pyriformis and minnow are within the safe range.

### 2.7. Larvicidal Activity

The compounds showed a range of toxicities against *Anopheles arabiensis* larvae, most mortalities being significantly higher than the negative control, acetone (*p* < 0.0001). Table 12 summarizes larval mortalities, which slightly increased, but not significantly (*p* > 0.10), from 24 to 48 h of exposure. Exposure to Temephos resulted in 100% mortality after 48 h. The most active compound besides the positive control was **6a**, resulting in over 93% larval mortality. Compounds **6d**, **6h,** and **6i** followed, killing 88%, 82% and 73% of the larvae, respectively, after 48 h of exposure.

## 3. Materials and Methods

### 3.1. General

Chemical reactants and solvents were obtained from Sigma-Aldrich (St, Louis, MO, USA) and used without modification. The melting point of the intermediate 4-hydrazinyl-8-(trifluoromethyl)quinolone (**5**) and target compounds (*E*)-4-(2-benzylidenehydrazinyl)-8-(trifluoromethyl)quinoline derivatives (**6a**–**i**) was determined by the open capillary method and was uncorrected. Reaction completion was monitored by the TLC method, using silica gel-coated aluminum sheets (Merck^60^F254 (Rahway, NJ, USA)) in a chloroform/methanol mixture. The spots were visualized using iodine vapors/ultraviolet light. The ^1^H-NMR, ^19^F-NMR, and ^13^C-NMR spectra were recorded on a Bruker (Mannheim, Germany) AMX-400 (400 MHz) spectrometer using dimethyl sulfoxide (DMSO-d_6_) as solvent and tetramethylsilane (TMS) as the internal standard. The splitting patterns are designated as follows: s, singlet; d, doublet; dd, doublet of doublet; t, triplet; m, multiplet. High-resolution mass spectra (HRMS) were recorded on Agilent (Santa Clara, CA, USA) Q-TOF-Mass Hunter instruments. The molecular ion peak (*m*/*z*) [M+1] corresponded to its molecular weight.

Chemicals and solvents used for the current project were 2-trifluoromethyl aniline, acrylic acid, hydroquinone, hydrazine hydrate, iodine, phosphorous oxychloride, ethanol, and aryl aldehydes such as 3-bromo-4-fluoro benzaldehyde, 2-methoxy-5-bromo benzaldehyde, 2-nitro-4,5-dimethoxy benzaldehyde, 3-bromo-4-methoxy benzaldehyde, 2-chloro-5-nitro benzaldehyde, 3-nitro-4-methoxy benzaldehyde, 2,4-dimethoxy benzaldehyde, 4-fluoro benzaldehyde, and 4-bromo benzaldehyde.

### 3.2. Synthetic Procedure of 4-Hydrazinyl-8-(trifluoromethyl)quinoline (***5***) [35]

The mixture of 4-chloro-8-trifluoromethyl quinoline (2.31 g, 1 equiv) and hydrazine hydrate (1.28 mL, 4 equiv) was refluxed in 50 mL of ethanol for 14–16 h. Reaction progress was monitored by TLC at suitable intervals, and upon the completion of the reaction, the solution was poured onto crushed ice; the precipitate was filtered, washed with ice-cold water, and dried. The compounds were recrystallized from methanol.

Appearance: Off-white; Yield: 92%; m.p. 144–145 °C; ^1^H-NMR (400 MHz, DMSO-d_6_, δ ppm) = 8.73 (s, 1H), 8.51 (d, *J* = 5.3 Hz, 1H), 8.43 (d, *J* = 7.9 Hz, 1H), 7.99 (d, *J* = 7.1 Hz, 1H), 7.47 (t, *J* = 7.8 Hz, 1H), 6.99 (d, *J* = 5.4 Hz, 1H), 4.50 (br s, 2H); MS: *m*/*z* = 227.18.

### 3.3. General Method for the Synthesis of (E)-4-(2-Benzylidenehydrazinyl)-8-(trifluoromethyl)quinolone (***6a***–***i***) [35]

A mixture of 4-hydrazinyl-8-(trifluoromethyl)quinoline (1 equiv) and substituted aromatic aldehydes (1 equiv) in 5 vol of ethanol was refluxed for 4–6 h. The reaction progress was monitored by TLC, and the resulting mixture was poured onto crushed ice. The brownish-colored solid was separated, filtered, and washed with cold water. The product was recrystallized from methanol.

#### 3.3.1. (*E*)-4-(2-(3-Bromo-4-fluorobenzylidene)hydrazinyl)-8-(trifluoromethyl)quinoline (**6a**)

Appearance: Pale yellow powder; ^1^H-NMR (DMSO-d_6_, δ ppm): 11.45 (s, 1H, -NH), 8.68 (d, 1H, *J* = 5.2Hz, trifluoromethyl quinoline), 8.62 (d, 1H, *J* = 8.4Hz, trifluoromethyl quinoline), 8.35 (s, 1H, =CH-), 8.15 (brd, 1H, *J* = 6.8 Hz, trifluoromethyl quinoline), 8.10 (brd, 1H, *J* = 7.2 Hz, trifluoromethyl quinoline), 7.85 (m, 1H, Ar-H), 7.65 (t, 1H, Ar-H), 7.55 (d, 1H, *J* = 5.2 Hz, trifluoromethyl quinoline), 7.45 (t, 1H, *J* = 8.4 Hz, Ar-H); ^19^F NMR (282 MHz, DMSO-d_6_): δ = −107.01 (s, 3F), −58.67 (s, 3F); ^13^C-NMR (DMSO-d_6_, δ ppm); 167.60, 157.82, 154.15, 150.81, 143.38, 134.17, 131.65, 129.66, 129.00, 128.02, 127.61, 124.65, 122.85, 117.80, 117.78, 113.77, 109.85; HRMS (ESI-TOF) (*m*/*z*): [M+1] calculated for C_17_H_10_BrF_4_N_3_, 412.0067; found, 412.0062.

#### 3.3.2. (*E*)-4-(2-(5-Bromo-2-methoxybenzylidene)hydrazinyl)-8-(trifluoromethyl)quinoline (**6b**)

Appearance: Yellow powder; ^1^H-NMR (DMSO-d_6_, δ ppm): 11.02 (br, 1H, -NH), 8.64 (d, 1H, *J* = 4.8 Hz, trifluoromethyl quinoline), 8.50 (d, 1H, *J* = 8.4 Hz, trifluoromethyl quinoline), 8.03 (d, 1H, *J* = 2.4 HZ, AR-H), 7.92 (d, 1H, *J* = 6.8 HZ, trifluoromethyl quinoline), 7.59 (s, 1H, =CH-), 7.44-7.42 (m, 2H, trifluoromethyl quinoline, Ar-H), 7.34 (d, 1H, *J* = 6.4 Hz, trifluoromethyl quinoline), 6.80 (d, 1H, *J* = 8.8 Hz, Ar-H), 3.82 (s, 3H, -OCH_3_); ^19^F NMR (282 MHz, DMSO-d_6_): δ = −58.67 (s, 3F); ^13^C-NMR (DMSO-d_6_, δ ppm); 157.12, 156.67, 153.93, 150.91, 143.12, 134.88, 132.07, 129.46, 128.12, 127.65, 124.27, 122.68, 119.00, 117.58, 113.58, 113.03, 109.51, 55.82; HRMS (ESI-TOF) (*m*/*z*): [M+1] calculated for C_18_H_13_BrF_3_N_3_O, 424.0267; found, 424.0262.

#### 3.3.3. (*E*)-4-(2-(4,5-Dimethoxy-2-nitrobenzylidene)hydrazinyl)-8-(trifluoromethyl)quinoline (**6c**)

Appearance: Pale orange powder; ^1^H-NMR (DMSO-d_6_, δ ppm): 11.64 (s, 1H, -NH), 8.97 (s, 1H, =CH-), 8.73 (d, 1H, *J* = 8.4 Hz, trifluoromethyl quinoline), 8.66 (d, 1H, *J* = 5.2 Hz, trifluoromethyl quinoline), 8.13 (d, 1H, *J* = 7.6 Hz, trifluoromethyl quinoline), 7.68–7.72 (m, 3H, Ar-H, trifluoromethyl quinoline), 7.59 (d, 1H, *J* = 5.2 Hz, trifluoromethyl quinoline), 4.35 (s, 3H –OCH_3_), 3.92 (s, 3H, -OCH_3_); ^19^F NMR (282 MHz, DMSO-d_6_): δ = −58.67 (s, 3F); ^13^C-NMR (DMSO-d_6_, δ ppm); 157.69, 156.08, 153.56, 152.01, 150.55, 143.61, 141.82, 129.32, 128.13, 127.78, 124.66, 122.43, 121.82, 117.56, 115.43, 113.45, 109.81, 56.58,56.58; HRMS (ESI-TOF) (*m*/*z*): [M+1] calculated for C_19_H_15_F_3_N_4_O_4_, 421.1118; found, 421.1111.

#### 3.3.4. (*E*)-4-(2-(3-Bromo-4-methoxybenzylidene)hydrazinyl)-8-(trifluoromethyl)quinoline (**6d**)

Appearance: Yellow powder; ^1^H-NMR (DMSO-d_6_, δ ppm): 11.31 (s, 1H, -NH), 8.623–8.686 [2H (d, 1H, *J* = 8.4 Hz, trifluoromethyl quinoline), (d, 1H, *J* = 5.2 Hz, trifluoromethyl quinoline)], 8.33 (s, 1H, =CH), 8.061–8.104 [2H (d, 1H, *J* = 2 Hz, Ar-H) (d, 1H, *J* = 7.2 Hz, trifluoromethyl quinoline)], 7.78 (d, 1H, *J* = 6.8 Hz, trifluoromethyl quinoline), 7.65 (m, 1H, Ar-H), 7.51 (d, 1H, *J* = 5.6 Hz, trifluoromethyl quinoline), 7.21 (d, 1H, *J* = 8.4 Hz, Ar-H), 3.92 (s, 3H, -OCH_3_); ^19^F NMR (282 MHz, DMSO-d_6_): δ = −58.67 (s, 3F); ^13^C-NMR (DMSO-d_6_, δ ppm); 159.61, 157.84, 153.83, 150.88, 143.51, 130.00, 129.33, 129.33, 128.13, 127.88, 127.51, 124.40, 122.83, 117.82, 113.81, 112.54, 112.43, 55.31; HRMS (ESI-TOF) (*m*/*z*): [M+1] calculated for C_18_H_13_BrF_3_N_3_O, 424.0266; found, 424.0262.

#### 3.3.5. (*E*)-4-(2-(2-Chloro-5-nitrobenzylidene)hydrazinyl)-8-(trifluoromethyl)quinoline (**6e**)

Appearance: Yellow powder; ^1^H-NMR (DMSO-d_6_, δ ppm): 11.70 (s, 1H, -NH), 8.80 (s, 3H, trifluoromethyl quinoline, Ar-H), 8.62 (d, 1H, *J* = 8.4 Hz, trifluoromethyl quinoline), 8.20 (dd, 1H, *J* = 2 Hz, *J* = 2.4 Hz, Ar-H), 8.13 (dd, 1H, *J* = 7.2 Hz, trifluoromethyl quinoline), 7.82 (d, 1H, *J* = 8.8 Hz, Ar-H), 7.69 (d, 1H, *J* = 8.8 Hz, Ar-H), 7.57 (d, 1H, *J* = 4.8 Hz, trifluoromethyl quinoline); ^19^F NMR (282 MHz, DMSO-d_6_): δ = −58.67 (s, 3F); ^13^C-NMR (DMSO-d_6_, δ ppm); 151.50, 146.76, 146.45, 144.77, 138.09, 137.34, 133.27, 131.38, 127.85, 127.78, 126.36, 126.17, 124.17, 123.41, 120.60, 117.39, 102.20; HRMS (ESI-TOF) (*m*/*z*): [M+1] calculated for C_17_H_10_ClF_3_N_4_O_2_, 395.0517; found, 395.0511.

#### 3.3.6. (*E*)-4-(2-(4-Methoxy-3-nitrobenzylidene)hydrazinyl)-8-(trifluoromethyl)quinoline (**6f**)

Appearance: Pale orange powder; ^1^H-NMR (DMSO-d_6_, δ ppm): 11.39 (s, 1H, -NH), 8.69 (dd, 1H, *J* = 4.8 Hz, trifluoromethyl quinoline), 8.61 (dd, 1H, *J* = 8.4 Hz, Ar-H), 8.37 (s, 1H), 8.29 (s, 1H, =CH), 8.07 (dd, 2H, *J* = 8.8 Hz, Ar-H), 7.63 (m, 1H, *J* = 7.6 Hz), 7.53 (dd, 1H, *J* = 5.2 Hz, trifluoromethyl quinoline), 7.42 (1H, dd, *J* = 8.8 Hz, Ar-H), 3.98 (s, 3H, -OCH_3_); ^19^F NMR (282 MHz, DMSO-d_6_): δ = −58.67 (s, 3F); ^13^C-NMR (DMSO-d_6_, δ ppm); 152.41, 151.40, 147.16, 144.76, 141.20, 139.68, 132.05, 127.83, 127.44, 126.75, 125.69, 123.17, 122.97, 122.57, 117.44, 114.63, 101.90, 56.90; HRMS (ESI-TOF) (*m*/*z*): [M+1] calculated for C_18_H_13_F_3_N_4_O_3_, 391.1011; found, 391.1006.

#### 3.3.7. (*E*)-4-(2-(2,4-Dimethoxybenzylidene)hydrazinyl)-8-(trifluoromethyl)quinoline (**6g**)

Appearance: Yellow powder; ^1^H-NMR (DMSO-d_6_, δ ppm): 10.22 (s, 1H, -NH), 8.72 (s, 1H, trifluoromethyl quinoline), 8.30 (s, 1H, =CH), 7.93 (m, 3H, *J* = 8 Hz, 8.8 Hz Ar-H), 7.42 (dd, 2H, *J* = 4.8 Hz, dd, *J* = 7.6 Hz), 6.52 (dd, 1H *J* = 2 Hz, *J* = 2.4 Hz, Ar-H), 6.39(dd, 1H, *J* = 2 Hz, Ar-H), 3.84 (s, 6H, -OCH_3_); ^19^F NMR (282 MHz, DMSO-d_6_): δ = −58.67 (s, 3F); ^13^C-NMR (DMSO-d_6_, δ ppm); 162.67, 159.09, 151.71, 146.70, 139.59, 127.62, 127.53, 125.65, 123.65, 123.57, 123.17, 122.93, 117.74, 115.74, 106.06, 102.66, 98.34, 55.63, 55.49; HRMS (ESI-TOF) (*m*/*z*): [M+1] calculated for C_19_H_16_F_3_N_3_O_2_, 376.1267; found, 376.1266.

#### 3.3.8. (*E*)-4-(2-(4-Fluorobenzylidene)hydrazinyl)-8-(trifluoromethyl)quinoline (**6h**)

Appearance: Off-white powder; ^1^H-NMR (DMSO-d_6_, δ ppm): 8.82 (s, 1H, -NH), 8.54 (s, 1H,), 8.01 (m, 3H, trifluoromethyl quinoline), 7.73 (dd, 2H, *J* = 5.6 Hz, *J* = 5.2 Hz, trifluoromethyl quinoline), 7.47 (dd, 2H, *J* = 7.2 Hz, Ar-H), 7.15 (dd, 2H, *J* = 8.8 Hz, *J* = 8.4 Hz, Ar-H); ^19^F NMR (282 MHz, DMSO-d_6_): δ = −111.37 (s, 3F), −58.67 (s, 3F); ^13^C-NMR (DMSO-d_6_, δ ppm); 165.08, 162.59, 145.23, 130.47, 128.89, 128.82, 127.89, 127.83, 127.79, 125.58, 123.47, 123.43, 122.87, 116.23, 116.12, 115.90, 102.98; HRMS (ESI-TOF) (*m*/*z*): [M+1] calculated for C_17_H_11_F_4_N_3_, 334.0951; found, 334.0961.

#### 3.3.9. (*E*)-4-(2-(4-Bromobenzylidene)hydrazinyl)-8-(trifluoromethyl)quinoline (**6i**)

Appearance: Pale yellow powder; ^1^H-NMR (DMSO-d_6_, δ ppm): 11.37 (s, 1H, -NH), 8.70 (d, 1H, *J* = 5.2 Hz, trifluoromethyl quinoline), 8.62 (dd, 1H, *J* = 8.4 Hz, trifluoromethyl quinoline), 8.37 (s, 1H, =CH), 8.10 (d, 1H, *J* = 6.4 Hz Ar-H), 7.74 (d, 2H, *J* = 6. 4 Hz, Ar-H, trifluoromethyl quinoline), 7.64 (dd, 3H, trifluoromethyl quinoline, Ar-H), 7.51 (dd, 1H, *J* = 4.8 Hz, trifluoromethyl quinoline); ^19^F NMR (282 MHz, DMSO-d_6_): δ = −58.67 (s, 3F); ^13^C-NMR (DMSO-d_6_, δ ppm); 151.48, 147.11, 146.98, 144.92, 142.32, 133.91, 131.78, 128.49, 127.88, 127.83, 126.71, 126.12, 125.84, 123.22, 122.58, 117.48, 101.90; HRMS (ESI-TOF) (*m*/*z*): [M+1] calculated for C_17_H_11_BrF_3_N_3_, 393.0161; found, 394.0156 [M+2].

### 3.4. Single-Crystal X-Ray Studies

A single crystal of the compound (**6d**), having colorless needle-shaped crystals with dimensions of 0.21 × 0.24 × 0.28 mm, was selected for XRD studies. Data were collected at 293 K with *MoKα* radiation having a wavelength of λ = 0.71073 Å. The intensity data set was examined using the SAINT software [42]; SHELXS and SHELXL [43,44] were used to solve the structure with direct methods and refined by a least-squares full-matrix method based on F2. Geometrical parameters were calculated using the PLATON [45]. ORTEP of the molecule and molecular packing visualization were plotted using MERCURY software [46].

#### Computational Studies

Gaussian 09 [47] software was used to study the DFT, the optimized structure, and electronic properties of the compound (**6d**), visualized with the Gaussian View 6 program [48]. The electronic properties (HOMO–LUMO energy) and MEP of the compound (**6d**) were calculated using the DFT/B3LYP method with the 6-311++G(d,p) basis set in the gas phase. Hirshfeld surface analysis and the 2D fingerprint plots were plotted using the Crystal Explorer 17.5 software [49,50]. In order to visualize the intermolecular interactions, Hirshfeld surface analysis is carried out, and also for the packing of the crystal.

Hirshfeld surfaces allowed the display of intermolecular interactions using distinct colors and intensities to depict short or long contacts and the relative strength of the interactions. It should be highlighted that the color scheme of a d_norm_ mapped on HS correlates to the strength of intermolecular interactions, which ranges from strong (red) to moderate (white) to weak (blue). The 2D fingerprint is a plot of d_e_ versus d_i_, where d_e_ and d_i_ are the distances from the HS to the nearest nucleus outside and inside the HS, respectively. It also provides an insight into crystal packing [51].

Crystal Explorer software 17.5 was used to study the building of the 3D energy frameworks with electron density wave function HF/3-21G [52], which revealed the topography of the intermolecular interactions in a crystal lattice. A 3D energy framework was performed using the standard red, green, and blue solid cylinders to visualize electrostatic (E_ele), dispersion (E_dis), and total energy (E_tot) components, respectively.

### 3.5. Computational Studies (Molecular Docking)

The title compounds 6a-i were studied through in silico investigation, interacting with the active site of acetylcholinesterase from the malaria vector, using docking simulations with AutoDock tools [53]. The main goal was to evaluate their binding affinity and gain insights into their potential as ligands for inhibiting the malaria vector. Initially, the ligands were individually drawn with ChemDraw V23.1.1 and then converted into a 3D format using Chem3D V23.1.1. The structures 6a-i were saved in .pdb format. These ligands were imported into AutoDock tools to perform docking simulations. The protein structure was obtained from the Protein Data Bank (https://www.rcsb.org/) with PDB ID: 6ARY. The protein was prepared by removing water molecules and existing ligands. Missing atoms were added to preserve structural integrity, and missing charges were included. Polar hydrogens and Kollman charges were added to ensure accurate protonation states for the docking. The docking results were visualized and analyzed with PyMol [54], while hydrophobic interactions in the protein–ligand complex were visualized using LIGPLOT+ software [55]. Finally, the docking poses were generated with PyMOL to create a 3D visualization of the binding interactions.

### 3.6. ADMET Prediction

Pharmacokinetic processes, also known as ADMET (Absorption, Distribution, Metabolism, Excretion, and Toxicity), are essential for understanding how drugs interact with the body and for predicting their pharmacodynamic effects. In silico ADMET predictions are crucial in optimizing potential lead compounds during drug discovery and development. In this study, we used the Swiss ADME web server [41] to predict the ADME parameters [1,2]. Additionally, the toxicity profile of the design compounds (**6a**–**i**) was also evaluated using the pkCSM web server [40], ensuring a comprehensive safety assessment of our target compounds (**6a**–**i**). SWISSADME [56], a free web tool, is used to explore the BOILED-Egg model [57] for compound **6a**–**i**.

### 3.7. Larvicidal Activity

The larvicidal activity was evaluated against *Anopheles arabiensis* larvae in accordance with the guidelines described by the WHO (1975) [58] under controlled insectary conditions simulating a malaria-endemic environment (temperature: 27.5 °C; humidity: 70%; and photoperiod: 12 h light/12 h dark). A test solution of final concentration of 4 µg/mL was prepared by dissolving 1 mg of the test compound in 1 mL of acetone (1 mg/mL), followed by dilution with 249 mL of distilled water. Thirty-third-instar larvae were introduced into each treatment container.

Negative controls were prepared using acetone and distilled water, while the positive control consisted of Temephos (4 µg/mL), a widely used emulsified organophosphate larvicidal agent used in malaria control programs. Larvae were fed a low-fat, specially formulated cat food throughout the assay. Larval mortality was recorded at 24 and 48 h post-exposure, and each treatment was conducted in triplicate. Mortality rates were calculated relative to the initial number of larvae introduced in each container.

### 3.8. Statistical Analysis

Generalized linear models using a quasi-binomial link function were used to assess differences in larval mortality between treatments and exposure times [59]. Larval mortality of *Anopheles arabiensis* was treated as the dependent variable, with the test compounds (test compounds **6a**–**i** and both controls) and the observation duration (24 h and 48 h) as fixed effects. Statistical significance was defined as a *p*-value < 0.05. The findings are displayed as the adjusted mean ± the standard error.

## 4. Conclusions

A novel series of benzylidenehydrazinyl derivatives of 8-trifluoromethyl quinolone (**6a**–**i**) was synthesized via a reaction of 4-hydrazinyl-8-(trifluoromethyl)quinoline with 3-bromo-4-methoxy benzaldehyde. The structural characterization of the synthesized compounds was performed using ^1^H-NMR, ^13^C-NMR, and mass spectrometry, and their antimalarial activity was subsequently evaluated. Among the series, compound **6a** has demonstrated notable larvicidal activity against the *Anopheles arabiensis* mosquito. Single-crystal XRD analysis revealed that the compound crystallized in a monoclinic lattice system with the space group P2_1_/c. Furthermore, theoretical geometry optimization carried out using DFT/B3LYP/6-311++G(d,p) showed excellent agreement with the experimentally determined bond lengths, bond angles, and torsion angles, supporting the structural integrity of the synthesized compounds. Further insights into the structural and electronic properties of the compound **6d** were obtained through DFT calculations, which provided an optimized molecular geometry. The DFT results highlighted the molecule’s high reactivity and revealed the key regions of MEP relevant to intermolecular interactions. The crystal packing was primarily stabilized by intermolecular interactions of the type N2-H2…N1 hydrogen bonding, as confirmed by Hirshfeld surface analysis mapped over various properties. Interaction energies and energy frameworks were also computed, revealing that dispersion energy frameworks E_dis were dominant stabilizing forces, surpassing classical electrostatic E_ele contributions, as visualized through three-dimensional interaction energy analysis. The most active compounds, **6a** and **6d**, also showed strong binding interactions with the target protein 6ARY and satisfactory ADMET properties. Thus, these compounds can be considered lead molecules for further optimization in the search for potent larvicidal agents.

## Data Availability

CCDC 2189866 contains the supplementary crystallographic data for this paper. These data are available on request via http://www.ccdc.cam.ac.uk/data_request/cif (accessed on 19 February 2025).

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
