# Peer review of "Synthesis of Novel Arylhydrazones Bearing 8-Trifluoromethyl Quinoline: Crystal Insights, Larvicidal Activity, ADMET Predictions, and Molecular Docking Studies"

_pharmaceuticals, 2025, doi:10.3390/ph18121804_

Round 1
Reviewer 1 Report
Comments and Suggestions for Authors
The work titled " Synthesis of Novel Arylhydrazones Bearing 8-Trifluoromethyl Quinoline: Crystal Insights and Growth Inhibition Activity against a Malaria Parasite" represents a the synthesis of novel antimalarial agents. The modifying quinoline structures with hydrazone groups was a great idea in a hybridization strategy method. This work reports the synthesis, structural characterization, and larvicidal testing of a new series of aryl hydrazones derivatives. According to the collected results compound 6a showed the highest larvicidal activity, causing significant mortality in Anopheles arabiensis larvae as well as the single-crystal XRD analysis of 6d revealed a monoclinic crystal system with space group P2₁/c, stabilized by N–H···N intermolecular hydrogen bonds. The findings of this work seem valuable but it was directed to physicochemical properties more than biological investigation and suitable contribution to be published in other journals since the scope is not fit to the pharmaceuticals journal. The work should include more biological evaluations tests, methods, safety and molecular docking data and analysis
Author Response
Comments and Suggestions for Authors
The work titled " Synthesis of Novel Arylhydrazones Bearing 8-Trifluoromethyl Quinoline: Crystal Insights and Growth Inhibition Activity against a Malaria Parasite" represents a the synthesis of novel antimalarial agents. The modifying quinoline structures with hydrazone groups was a great idea in a hybridization strategy method. This work reports the synthesis, structural characterization, and larvicidal testing of a new series of aryl hydrazones derivatives. According to the collected results compound 6a showed the highest larvicidal activity, causing significant mortality in Anopheles arabiensis larvae as well as the single-crystal XRD analysis of 6d revealed a monoclinic crystal system with space group P2₁/c, stabilized by N–H···N intermolecular hydrogen bonds. The findings of this work seem valuable but it was directed to physicochemical properties more than biological investigation and suitable contribution to be published in other journals since the scope is not fit to the pharmaceuticals journal. The work should include more biological evaluations tests, methods, safety and molecular docking data and analysis.
Reply: We sincerely thank the reviewer for their valuable comments and constructive suggestions to improve the quality of our manuscript. In accordance with the recommendations, we have conducted computational studies and incorporated the results into the revised manuscript (Tables 10 and 11, Figure 12, and Table S1). Additionally, ADMET analysis, along with detailed molecular docking data and interpretation, has been included to strengthen the biological evaluation section and support the pharmacological relevance of the synthesized compounds. We believe these additions substantially enhance the scientific depth and alignment of the manuscript with the journal’s scope.

Reviewer 2 Report
Comments and Suggestions for Authors
The present manuscript reports synthesis of nine novel arylhydrazones, containing quinoline core; characterization of one of the hydrazones via single-crystal X-ray diffraction and a set of computational methods, and screening of larvicidal activity against Anopheles arabiensis larvae for the obtained compounds.
Unfortunately, on my opinion, the main subject of the article – the investigation of crystal and molecular structure of compound 6d – is not connected with pharmaceutical or related applications. No data on larvicidal activity are given for this very compound (see Table 10). And even if such activity has been reported, there is no point to thoroughly investigate the molecular structure without any mechanism of action and molecular targets discussed.
Therefore, I cannot recommend accepting the manuscript for publication in Pharmaceuticals.
Nevertheless, I would recommend submitting the article to a journal, focused on synthesis and molecular structure of compounds.
Besides, addressing the following issues may improve the paper quality.
Major issues:
- Elemental analysis or HRMS data should be provided for novel compounds.
Minor issues:
- In the title “growth inhibition activity” should be replaced for “larvicidal activity”.
- A closely related article has been previously published to describe the compounds differing from those reported in the present work only with substituents in aromatic moieties: Monatsh. Chem. 2015 146, 2041–2052, DOI: 10.1007/s00706-015-1570-0. It should be cited and the obtained results, including X-ray analysis data, should be compared.
- Discussing the importance of fluorine for drug design, it would be better to cite a corresponding review work, e.g., J. Med. Chem. 2015, 58, 8315−8359, DOI: 10.1021/acs.jmedchem.5b00258.
- It would be useful to provide also the therapeutic action of the imaged drugs In Figure 1.
- It would be better to move Scheme 1, illustrating the discussed compounds, to the beginning of Section 2.1.
- Literature reference should be given for previously described compound 5.
- In descriptions of synthetic procedures amounts of reagents and solvents (g or mL) should be given.
- For all fluorine-containing compounds 13C NMR spectra should be described with respect to fluorine-carbon spin-spin interaction. Signals, corresponding to CF3 group, should be described as quartets with 1JCF approx. 270 Hz, and so far for more distant nuclei.
- For all fluorine-containing compounds 19F NMR spectra should be registered.
Author Response
Comments and Suggestions for Authors
The present manuscript reports synthesis of nine novel arylhydrazones, containing quinoline core; characterization of one of the hydrazones via single-crystal X-ray diffraction and a set of computational methods, and screening of larvicidal activity against Anopheles arabiensis larvae for the obtained compounds.
Unfortunately, on my opinion, the main subject of the article – the investigation of crystal and molecular structure of compound 6d – is not connected with pharmaceutical or related applications. No data on larvicidal activity are given for this very compound (see Table 10). And even if such activity has been reported, there is no point to thoroughly investigate the molecular structure without any mechanism of action and molecular targets discussed.
Therefore, I cannot recommend accepting the manuscript for publication in Pharmaceuticals.
Nevertheless, I would recommend submitting the article to a journal, focused on synthesis and molecular structure of compounds.
Besides, addressing the following issues may improve the paper quality.
Major issues:
- Elemental analysis or HRMS data should be provided for novel compounds.
Reply: We thank the reviewer for their valuable comments, which helped us improve our manuscript to meet high standards. In the original submission, we added LC-MS data recorded on a Perkin-Elmer 018444Y, a triple quadrupole LC-MS spectrometer, and on a Waters Xevo TQD with MassLynx software. To address the reviewer’s comments and improve our manuscript, we have added high-resolution mass spectra (HRMS) recorded on Agilent Q-TOF MassHunter instruments to the ESI file, Page numbers 7, 11, 15, 19, 23, 27, 31, 35, and 39, and the values have been updated accordingly in the revised manuscript, page numbers 3 and 4.
Minor issues:
- In the title “growth inhibition activity” should be replaced for “larvicidal activity”.
Reply: It has been updated in the revised manuscript and ESI file as well.
- A closely related article has been previously published to describe the compounds differing from those reported in the present work only with substituents in aromatic moieties: Monatsh. Chem. 2015 146, 2041–2052, DOI: 10.1007/s00706-015-1570-0. It should be cited and the obtained results, including X-ray analysis data, should be compared.
Reply: The article DOI: 10.1007/s00706-015-1570-0 is cited in the manuscript, and X-ray analysis data are compared and updated in the revised manuscript on page number 26.
- Discussing the importance of fluorine for drug design, it would be better to cite a corresponding review work, e.g., J. Med. Chem. 2015, 58, 8315−8359, DOI: 10.1021/acs.jmedchem.5b00258.
Reply: Reference https://pubs.acs.org/doi/10.1021/acs.jmedchem.5b00258 has been cited on page 2.
- It would be useful to provide also the therapeutic action of the imaged drugs In Figure 1.
Reply: In the updated manuscript, the names of the six active pharmaceutical ingredients, their structures, and uses are added in Figure 1 on page number 3.
- It would be better to move Scheme 1, illustrating the discussed compounds, to the beginning of Section 2.1.
Reply: As the reviewer recommended, we feel Scheme 1 is appropriate to include in Section 2.1, and accordingly, we moved it for better presentation on page number 4.
- Literature reference should be given for previously described compound 5.
Reply: Literature reference is updated in the revised manuscript on pages 4, 7, and 18.
- In descriptions of synthetic procedures amounts of reagents and solvents (g or mL) should be given.
Reply: The stoichiometric equivalence of reactants and solvent is provided on page 18 for the specific synthesis of intermediate 5, and the title compounds 6a-i are presented in a general stoichiometric equivalence form.
- For all fluorine-containing compounds 13C NMR spectra should be described with respect to fluorine-carbon spin-spin interaction. Signals, corresponding to CF3 group, should be described as quartets with 1JCF approx. 270 Hz, and so far for more distant nuclei.
Reply: 19F NMR spectra of the title compounds 6a-i have been recorded on a Bruker AMX-400 (400 MHz) spectrometer using dimethyl sulfoxide (DMSO-d6) as solvent and tetramethylsilane (TMS) as the internal standard, and the spectra are updated in the ESI file on pages 5, 9, 13, 17, 21, 25, 29, 33, and 37. The 19F NMR values are also updated in the revised manuscript from pages 2-4.
- For all fluorine-containing compounds, 19F NMR spectra should be registered.
Reply: 19F NMR spectra of the title compounds 6a-i have been recorded, and the spectra are updated in the ESI file on pages 5, 9, 13, 17, 21, 25, 29, 33, and 37. The 19F NMR values are also updated in the revised manuscript from pages 2-4.

Round 2
Reviewer 1 Report
Comments and Suggestions for Authors
The authors were tried to improve their work well by adding molecular docking works, and they should :
1-Protein code in table 10 should be written as PDB : 6ARY
2-the type of binding interactions should be added to table 10 accordingly
3-the 3D protein style for compound 6h should be like the other compounds
4- it is recommended to add the boiled egg for ADME which could be extracted from Swiss ADME site
5- the title could be incloud the ADME and molecular docking words to be more attractive for readers
6- the aims of this study could be improved in the last paragraph of introduction
7-in the NMR data they should add the J values for dd as two value not one
8-all chemicals and reagents used in this study should be added in separate section in the methods
9-the conclusion should be improved too with main finding and without listing of any values
Author Response
Comments and Suggestions for Authors
The authors were tried to improve their work well by adding molecular docking works, and they should :
We sincerely thank the reviewer for providing important comments to improve our manuscript to the best possible form.
1-Protein code in table 10 should be written as PDB : 6ARY
Reply: The changes have been incorporated accordingly into Table 10.
2-the type of binding interactions should be added to table 10 accordingly
Reply: The binding interactions are listed in Table 10 accordingly.
3-the 3D protein style for compound 6h should be like the other compounds
Reply: The changes have been incorporated into Table 10.
4- it is recommended to add the boiled egg for ADME which could be extracted from Swiss ADME site
Reply: BOILED-Egg studies are included in the main manuscript.
5- the title could be include the ADME and molecular docking words to be more attractive for readers
Reply: As per the reviewer’s suggestion, we have updated our title to “Synthesis of Novel Arylhydrazones Bearing 8-Trifluoromethyl Quinoline: Crystal Insights, Larvicidal Activity, ADMET Predictions, and Molecular Docking Studies.”
6- the aims of this study could be improved in the last paragraph of the introduction
Reply: AS per the reviewer's suggestion, we have included the aim of this study in the last paragraph of the introduction as “The synthesis of quinoline derivatives has gained significant interest in organic chemistry because of their biological and pharmacological uses mentioned earlier. Based on these findings, this study reports the synthesis of benzylidenehydrazinyl derivatives of 8-trifluoromethyl quinolone, characterized by ¹H NMR, 19F NMR, ¹³C NMR, high-resolution mass spectrometry (HRMS), X-ray crystallography, Hirshfeld surface analysis, and density functional theory (DFT) studies, aimed at developing new larvicidal agents.”
7-in the NMR data they should add the J values for dd as two value not one
Reply: In the revised J values for dd, on the page number 22 has been added.
8-all chemicals and reagents used in this study should be added in separate section in the methods
Reply: In the revised section, the chemical names are separated from 3. Experimental Studies/3.1. General, on page number 19, and added at the end of 3.1. General on page number 20.
9-the conclusion should be improved too with main finding and without listing of any values
Reply: A significant crystallography conclusion with certain values has been removed to reach an effective conclusion.

Reviewer 2 Report
Comments and Suggestions for Authors
I appreciate the authors' attentions to the recommendations considering the work. As most of the remarks have been adressed, I can recommend to accept the manuscript for publication.
Following issues still should be considered:
- While describing HRMS data it would be better to give calculated [M+H]+ monoisotopic (or exact) mass values, necessarily, taking into account the charge. For 6a it will be 412.0067, that makes excellent correspondence to found value 412.0062.
- The question remains, whether the quartets corresponding to CF3 groups were observed in 13C NMR spectra. If yes, they should be described as quartets with corresponding spin-spin coupling constants. If no, the explanation should be given: was it due to low concentrations of the samples, or, may be 13C NMR spectra were obtained in fluorine-decoupled mode?
- In section 3.1 standard for 19F NMR should be given.
Author Response
Comments and Suggestions for Authors
I appreciate the authors' attentions to the recommendations considering the work. As most of the remarks have been adressed, I can recommend to accept the manuscript for publication.
Reply: We appreciate the reviewer's patience and time in providing all constructive critical comments to improve our manuscript to the best possible form.
Following issues still should be considered:
- While describing HRMS data it would be better to give calculated [M+H]+ monoisotopic (or exact) mass values, necessarily, taking into account the charge. For 6a it will be 412.0067, that makes excellent correspondence to found value 412.0062.
Reply: We thank the reviewer for their valuable comment. The HRMS data has been revised, retaining monoisotopic (or exact) mass values and is now shown on pages 21 and 22.
- The question remains, whether the quartets corresponding to CF3 groups were observed in 13C NMR spectra. If yes, they should be described as quartets with corresponding spin-spin coupling constants. If no, the explanation should be given: was it due to low concentrations of the samples, or, may be 13C NMR spectra were obtained in fluorine-decoupled mode?
Reply: We thank the reviewer for this valuable observation. The expected quartets for the CF₃ group were not visible in the 13C NMR spectrum. This is because the spectra were acquired in 19F decoupled mode, following to the default settings of the NMR instrument used, which suppresses the 13C - 9F coupling and produces singlet signals. This has now been clarified in the revised manuscript.
- In section 3.1 standard for 19F NMR should be given.
Reply: In section 3.1, the 19F NMR standard is listed and highlighted in green on page 19.
